# Do Charity or Non-Charity Sporting Events Have a Greater Influence on Participants' Warm Glow?: An Experimental Survey

**Watchara Chiengkul, Patcharaporn Mahasuweerachai ***  and Chompoonut Suttikun

Faculty of Business Administration and Accountancy, Khon Kaen University, Khon Kaen 40002, Thailand
* Correspondence: patchaporn@kku.ac.th

**Abstract:** Although the literature shows that consuming pro-social products increases warm glow, a psychological factor that contributes to consumer experience and satisfaction, it is unclear whether participating in a charity sporting event results in the same outcome. This research addresses this gap by testing the effects of participating in charity running events and altruism on the warm glow of participants. A scenario-based survey was employed to collect data from 180 respondents who had previously participated in charity running events. Multiple regression analysis results showed that participating in a charity (vs. non-charity) running event increased participants' warm glow. Altruism also had a significant positive impact on their warm glow; however, this effect was independent of the percentage of proceeds donated by the event to charity.

**Keywords:** warm glow; altruism; charity sporting events; running events; tourism

## 1. Introduction

Sporting events have gained increasing attention globally and become a part of tourism due to the unique experiences they offer [1]. With a global growth rate of more than 49% from 2008 to 2018, running has become a popular activity. This is particularly true in Asia, which has the highest growth rate at more than 262% [2]. In 2019, the new coronavirus (COVID-19) posed enormous social, economic, and political problems. Since the coronavirus outbreak, every element of an individual's life appears to have undergone a significant transformation, and numerous businesses have had to deal with massive sales losses [3]. In terms of event organization, local and international sporting events have been one of the sectors hardest hit by the crisis and have been postponed, canceled, or changed [4]. Currently, the situation is improving, as numerous sporting events are starting to restart in line with the "new normal". Therefore, it is important to concentrate on how the charity running event sector responds to the opportunities and challenges to reshape the industry toward sustainability in the future.

Charity running events have different goals compared to marathons (non-charity running events). A charity running event is a fund-raising activity aimed at raising awareness of and money for social issues, such as education, medicine, poverty, or the environment. Participants are required to pay registration fees and all or some of the proceeds are donated towards these social issues [5–7]. Such events combine fund-raising and physical activities to provide benefits for both participants and society [7,8]. Individuals can become involved in a charity running event in numerous ways, such as by volunteering, purchasing event memorabilia, making a charitable contribution, or registering as a participant [9]. Doing good things for others, such as facilitating various forms of donations [10–12], leads to a sense of happiness through giving to others. This feeling, referred to as "warm glow", is explained by the warm glow of giving theory [13]. An individual who has an intrinsic impulse to engage in socially beneficial activities experiences a warm glow. Experiencing

one's own happiness and that of others reflects the moral values of people who consider their own interests and those of others [14].

Human beings' social contribution behavior stems from altruistic factors, which are fundamental human values that guide people to live socially acceptable lives [15,16] and promote warm glow. Altruism involves an individual's selfless concern for others, often expressed through behavior of giving to or helping others to their benefit [17]. Hartmann et al.'s [14] research has confirmed that altruistic personal traits are what causes warm glow when individuals engage in charitable activities [18]. Previous research has provided evidence that the consumption of environmentally friendly products [14,19–22] also creates a warm glow feeling, enhancing the positive aspects of a consumption experience. However, the effect of attending charity running events on warm glow remains unclear.

To our knowledge, no research has yet explored how emotions and feelings related to altruism and warm glow drive individuals to participate in charity running events and to enjoy them as long-term participation experiences. The primary goal of organizing a charity run focuses on contributing to social issues in the community by creating an activity to attract altruistic people. It is, therefore, reasonable to expect that, in addition to altruism, participating in charity running events might induce participants to feel a warm glow. Thus, the main purpose of the current study is to reach a deeper understanding of the factors influencing warm glow. In addition, it aims to determine whether participants experience a warm glow feeling differently and, if so, whether this is related to the amount of donations (partial or full) contributed to the charity run. Accordingly, the researchers formulated the research questions as follows:

(1) Can participating in charity running events cause warm glow?

(2) Can altruism cause warm glow?

(3) How does the interaction between altruism and types of running event affect warm glow?

This study provides greater insight into charity event participation experiences, as well as having implications for relevant theory.

## 2. Literature Review and Hypotheses Development

### 2.1. Charity Sporting Events and Warm Glow

The concept of warm glow has been used to describe human behaviors related to social contributions in a variety of contexts [14,23–25]. A review of related research showed that warm glow is related to social behavior, such as charity donation [26], blood donation [12,27], giving money to others [11], purchasing products related to social marketing [28,29], volunteering in an environmental project [30], buying environmentally friendly products [14,19–22,31], and participating in social responsibility activities [10,32]. Tezer and Bodur [33] determined that the use of environmentally friendly products induced a warm glow in participants and created an enjoyable experience [30]. In the context of charity running events, Filo et al. [34] interviewed participants about their experiences and found that attending charity sporting events stimulated positive feelings in them related to engaging with people, overcoming physical limitations, and making social contributions, enabling participants to enjoy the experience. In addition to increasing their awareness of social problems and making contributions by attending charity sporting events, participants also gained a meaningful experience from doing so [35]. Nonetheless, no empirical evidence exists to elucidate the relationship between warm glow and involvement in charity running events. Thus, this study investigated the relationship between the two.

Recent research has shown that the consumption of products or services that are environmentally or societally friendly and participation in charity activities can lead consumers to experience warm glow. This feeling impacts the overall product and experience evaluation [14,30,33,34]. Therefore, it can be assumed that participating in charity running events would lead to a feeling of warm glow. This assumption led to our first hypothesis 1 (H1):

**Hypothesis 1 (H1).** *Compared with non-charity running events, charity running events result in greater warm glow feelings.*

*2.2. Altruism and Warm Glow*

Altruism is about doing good and should lead to feeling good. It involves the capacity to help others in need [36,37] and can be defined as an emotional expression of compassion, development, understanding, and care for others or society without consideration of benefits to the self [17]. Rushton et al. [38] used the term "altruistic personality" to describe these character traits, and Hartmann, Eisend, Apaolaza, and D'Souza [14] noted that these particular behavioral patterns reflect altruism. Altruism can also be seen in the human values that guide the way people live to gain societal acceptance [15,16]. This desire to gain acceptance is one driving force in the motivation of pro-social behavior [39].

According to several empirical studies, people who engage in pro-environmental activities promote altruistic ideals (e.g., [15,16,18,40]). Van der Linden [41] found that altruistic behavior was affected by moral norms and played a significant role in charitable intentions. Ferguson, Taylor, Keatley, Flynn, and Lawrence [12] found that the giving of blood donations was not only driven by the donor's altruistic motivation, but also by self-regarding motivations, such as warm glow [42]. Moreover, Hartmann, Eisend, Apaolaza, and D'Souza [14] found that altruistic personality traits may generate warm glow when people engage in environmentally beneficial behavior, where warm glow is the main driver of that behavior [18]. Several studies have confirmed that altruistic personality traits lead to warm glow [18,26,43]. According to the warm glow theory, human charitable conduct may be "impurely altruistic", which refers to the simultaneous maintenance of both altruistic and egoistic (selfish) incentives for giving. In other words, humans can also have reciprocity expectations, such as considerations of fame or social standing [26]. It can be concluded that altruism enhances warm glow and connects emotional expression to pro-environmental behavior, which led to Hypothesis 2 (H2):

**Hypothesis 2 (H2).** *Altruism results in warm glow.*

As noted, previous research has examined the relationship between charity activities and warm glow, or altruism and warm glow separately. How charity running events influence warm glow when participants have different levels of altruism has not yet been explored. It is possible that altruism may have a more significant impact on participating in charity running events that donate all or some money to charity, resulting in warm glow, irrespective of the amount of donations made [26]. Thus, this study proposed a third hypothesis (H3):

**Hypothesis 3 (H3).** *There is an interaction between altruism and types of running events, such that the effect of altruism on warm glow in relation to a charity running event increases compared to a non-charity running event.*

**3. Materials and Methods**

*3.1. Study Design and Instrument*

This study employed a scenario-based survey to test whether attending a charity running event increases warm glow and whether altruism can enhance this effect. The scenario-based design was considered appropriate for this study as the variable of interest could be manipulated, thus increasing internal validity [44]. Three scenarios were designed to represent different types of running event: Scenario 1 (SC1) described a non-charity running event where all the money from registration fees went to the host organizer. Scenario 2 (SC2) described a charity running event in which all proceeds from registration fees were donated to charity. Scenario 3 (SC3) described a charity running event in which a portion of the registration fees was donated to charity. Each scenario provided identical information except for a slightly altered last sentence to test the effect of event type (presented

in Appendix A). Each participant received only one scenario description. The content of the three scenarios was reviewed and validated by three academic professors who had experience in both charity and non-charity running events.

Data was collected through a self-administered questionnaire composed of four sections. To verify that the respondents met the sampling criteria, the first section contained two screening questions: "Have you ever participated in a charity run?" and "When did you last participate in a charity run?". Respondents could complete the next section of the questionnaire if they answered yes to the first questions and if they had participated in a charity run within a few years. If they responded no, they did not participate further. The second section provided a written scenario and manipulation check questions. The third section comprised questions to measure warm glow and altruism. The last section contained demographic information questions.

### 3.2. Sample and Data Collection

The study participants were selected based on a non-probability sampling method—convenience sampling [45]. Because the true population is unknown and no complete sample frame exists, probability-based sample unit selection could not be used. Additionally, concerns about the convenience of the situation were considered [46]. Due to the COVID-19 pandemic, most charity running organizers had postponed and suspended their events. Therefore, the researchers used questionnaires to collect data from a sample of experienced runners participating in running events at a public community park instead. Consequently, convenience sampling was suitable for this study. Participants comprised Thai runners at the most popular public park in Khon Kaen, Thailand who were over 19 years of age and had participated in at least one charity running event organized in Thailand. Before the coronavirus outbreak, charity running was a popular sporting event in Thailand. On average, more than 700 events were held each year, and the trend has continued to grow [47]. As the central location of government agencies and state enterprises in the region, Khon Kaen is one of the provinces in Thailand that hosts numerous charity running events. In addition, it houses the consulates of the People's Republic of China, Vietnam, Laos, and Peru, as well as various educational institutions, hospitals, and medical centers. It has also been selected as one of the Meetings, Incentives, Conferences, and Exhibitions (MICE) destinations that the government supports for various events; thus, relevant agencies organize various charity running events to raise funds to help with various issues [48].

A total of 195 questionnaires were generated covering scenarios 1 to 3. The questionnaire (65 for each scenario) was randomly distributed to runners as researchers encountered them at the park. Once they passed the screening questions, each respondent was given a questionnaire to complete. The respondents were recruited from December 1 to 31 December 2020. All were collected and checked for completeness. After eliminating incomplete questionnaires, a sample of 180 responses was retained for analysis (92.31% response rate).

### 3.3. Variables and Measurements

The dependent variable in this study was warm glow, which was measured using four items (Cronbach's alpha coefficient of 0.982) developed based on previous studies [10,22,32]. The items included (1) I feel a warm glow from helping others through this running event, (2) I feel self-satisfied from helping others through this running event, (3) I feel delight from helping others through this running event, and (4) I feel good from helping others through this running event.

Independent variables tested in this analysis were types of running events and altruism. The types of running event were manipulated using three scenarios, including SC1, SC2, and SC3; thus, they were coded as dummy variables. Altruism was measured by five questions (Cronbach's alpha coefficient of 0.832) adapted from Hartmann, Eisend, Apaolaza, and D'Souza [14]. The items included (1) If I get the chance, I will donate to charities or environmental organizations; (2) If I get the chance, I will donate to underprivileged or

needy people; (3) If I get the chance, I will work as a volunteer for a charity or environmental organization; (4) If I get the chance, I will willingly let others get served first when they are in urgent need; and (5) If I get the chance, I will willingly give up my seat on public transport to others. The warm glow and altruism variables were measured using a seven-point Likert scale ranging from (1) totally disagree to (7) totally agree.

### 3.4. Data Analysis

A total of 195 questionnaires were dispensed and returned. After deleting incomplete responses, 180 questionnaires remained for analysis. Data was analyzed using SPSS version 23 software. Descriptive statistics were used to describe the respondents' characteristics. Multiple regression was employed to test the hypotheses. Regression modeling is a suitable method for predicting the variables that influence warm glow. The most important elements, those that can be ignored, and the relationships between them can be confidently established [49]. The model for analysis is presented below:

$$\text{Warm glow} = \alpha_0 + \beta_1 SC2 + \beta_2 SC3 + \beta_3 \text{Altruism} + \beta_4 SC2 \times \text{Altruism} + \beta_5 SC2 \times \text{Altruism} + \varepsilon$$

## 4. Results

### 4.1. Respondents' Profile

Most respondents were between 19 and 29 years of age (53.53%). In relation to gender, 41.7% were male, 48.3% were female, and 10.0% were LGBTQ+. The respondents comprised mostly students (33.3%). Approximately 46.7% of the respondents had a monthly income of 10,001–20,000 Baht. Most of the respondents participated in marathons one or two times per year (71.7%) and participated in charity running events one or two times per year (90.5%).

### 4.2. Hypotheses Testing Results

To test the effect of type of running event on warm glow, as well as the interaction effects of the independent variables on warm glow, multiple regression was performed. The results revealed that the regression model was significant ($F_{(5, 179)} = 774.124$, $p < 0.001$) and $R^2$ was 0.957. That is, the independent variables in the model could explain 96% of the variance in warm glow.

The regression results presented in Table 1 show that a particular type of running event had positive effects on warm glow, supporting H1. Specifically, respondents in the SC2 ($b = 0.569$; $t = 2.961$; $p < 0.01$) and SC3 ($b = 0.446$; $t = 3.122$; $p < 0.01$) groups reported a significantly higher warm glow than those in the reference group (SC1). That is, participating in a charity running event that donates all proceeds to charity (SC2) and participating in a charity running event that donates some of the proceeds to charity (SC3) resulted in greater warm glow compared to participating in a non-charity running event (SC1). In addition, the magnitude of SC2's impact on warm glow was greater than that of SC3.

The results also support H2. Altruism had a positive and significant impact on warm glow ($b = 0.065$; $t = 2.344$; $p < 0.05$). That is, the more altruistically a person behaves, the greater their warm glow. Furthermore, the results show that the interaction terms, SC2 × Altruism ($b = 0.042$; $t = 2.344$; $p < 0.05$) and SC3 × Altruism ($b = 0.048$; $t = 2.344$; $p < 0.01$) had significantly positive effects on warm glow, supporting H3. Specifically, the impact of altruism on warm glow was significantly greater when participating in charity running events. However, the magnitude of altruism's effect on warm glow in SC2 was very similar to that in SC3. That is, at the same level of altruism, SC2's effect on warm glow did not differ from that of SC3.

**Table 1.** Results of multiple regression.

| Variable | Dependent Variable: Warm Glow | | | | |
|---|---|---|---|---|---|
| | b | Std. Error | β | t-Value | *p* |
| SC2 | 2.758 | 0.932 | 0.569 | 2.961 | 0.003 ** |
| SC3 | 2.162 | 0.692 | 0.446 | 3.122 | 0.002 ** |
| (Reference group: SIC) | | | | | |
| Altruism | 0.191 | 0.082 | 0.065 | 2.344 | 0.020 * |
| SC2 × Altruism | 0.311 | 0.147 | 0.420 | 2.108 | 0.036 * |
| SC3 × Altruism | 0.413 | 0.122 | 0.480 | 3.397 | 0.001 ** |
| (Reference group: SC1 × Altruism) | | | | | |
| $R^2$ = 0.957, Adjusted $R^2$ = 0.956 | | | | | |
| F = 774.124, *p* < 0.001 | | | | | |

Notes: b = Unstandardized coefficients, β = Standardized coefficients, ** *p* < 0.01, * *p* < 0.05.

## 5. Discussion and Conclusions

The results of this study show that participating in a charity running event that donates all proceeds to charity (SC2) and participating in a charity running event that donates some of the proceeds to charity (SC3) resulted in a greater warm glow than participating in a non-charity running event (SC1). In addition, the magnitude of SC2's impact on warm glow was greater than that of SC3. The findings of this study align with the participants' goals of participating in charity running events for self-reflection through empathy, self-image enhancement, self-worth performance, and a desire to improve charities [50]. These differ from participating in non-charity running events for recreation, intellectual enhancement, social connections between participants, physical enhancement, and success in competition. People who attend charity running events tend to experience a higher level of warm glow than people who join non-charity running events. This phenomenon may be explained by the theory of the warm glow of giving, which states that the perception of self-worth gained through the feeling of being a giver causes an inner warm glow [26]. Accordingly, warm glow is the key factor that makes participants have meaningful experiences when participating in charity sporting events. It is suggested that if a person has set goals to help with social issues and they know that the donations from an event have been used to support the issues, they will feel more satisfied because it confirms that the good deed has been successfully undertaken [14,51]. It also results in participants experiencing increased warm glow [52].

Furthermore, this study demonstrates that altruism is a critical factor in people experiencing warm glow. This has been identified in other studies as the good feeling of being a giver [17,36,37], which provides a guideline for living in a society [15,16]. Similar to the study of Van der Linden [41], it was found that the motivation to develop pro-social behavior arises from altruism [12], which leads to a warm glow feeling that results in further willingness to help others [14,18,26,43].

Perhaps not surprisingly, the results indicated that the impact of altruism on warm glow was significantly greater when participating in charity running events. Nevertheless, surprisingly, the magnitude of altruism's effect on warm glow in participating in a charity running event that donates all proceeds to charity (SC2×Altruism) was very similar to that of participating in a charity running event that donates only some of the proceeds to charity (SC3×Altruism). That is, at the same level of altruism, the increase in warm glow from SC2×Altruism was similar to that from SC3×Altruism. Therefore, the results of this study suggest that an altruistic person who knows that their donation, whether all or part, goes to charity, believes that they are doing good for others, resulting in emotional satisfaction [14].

The results of this study suggest that it is important for organizers to build trust and goodwill among the participants of their events. Clearly conveying the objectives of the event and maintaining transparency in the allocation of donations allows participants to have a good experience of the event [14,51,52]. The event will be successful as a result, and the business will be more sustainable. Hence, charity running events can be considered a

form of activity that allows participants to perceive their own self-worth by doing good for others and results in the occurrence of inner warm glow among participants [26,34]. Moreover, if humans maintain altruistic personality traits as a basis for their lives, they are more likely to be motivated to do good [53].

## 6. Implications, Limitations, and Future Research

The results of the study are consistent with the theory of the warm glow of giving. The participants' feeling of warm glow from the charity running event was the same irrespective of the amount of donations given to charity. Through experiment and model investigation, this study contributes to understanding and expanding the body of knowledge with respect to the implications of the warm glow theory for charity running events. Previous studies that investigated the warm glow of giving theory focused on products (e.g., [14,19–22,28–31]), or the context of hospitality (specifically, hotels, restaurants, and airlines) [30,54–57]. No previous research had been conducted in the context of sporting events. Therefore, this study's research integrated understanding of warm glow derived from the product context with the event and hospitality context. The enhancement and extension of this body of knowledge using a scenario-based investigation provides deeper understanding of the factors that influence the feeling of warm glow in the context of charity running events.

In terms of application, government agencies can use this knowledge to formulate policies and organize charity running events that bring the greatest benefits to social issues. Such policies may include supporting events, acting in cooperation with relevant parties, and raising awareness of the value of doing good for others and the resulting warm glow experienced by participating. This could be facilitated by creating a documentary to educate people about social issues to increase their awareness and encourage them to maintain personal norms of being good citizens who actively participate in preventing and solving social problems. Alternatively, government agencies might promote family institutions that encourage family members to participate in helping society, such as by joining a charity running event that can also benefit physical and mental health [58]. Additionally, organizers of charity events should express their intentions clearly and be credible, accountable, transparent, and ethical in managing donations to contribute to their targeted social causes [52]. Furthermore, the idea that people who participate in charity sporting events are contributing to society can be promoted [59]. Lastly, it is important to help people realize that they can contribute to society and create positive social change [60,61], with the consequence that they will experience warm glow associated with having behaved in a socially and environmentally responsible manner.

The main limitation of the present study is that the participants were largely young people under the age of 29. Although this was the study's primary target group, it would be interesting to investigate the behaviors and perspectives of other age groups. Moreover, the study focused on Thailand and it would also be interesting to replicate this study in other countries. Future research could broaden the study sample to include different age groups and nations to compare whether differences exist in the levels of warm glow experienced. Future research should also investigate other factors that may affect warm glow and the results of feeling warm glow, such as perceived social value. This is a further factor that can affect the feeling of warm glow and may lead to an increased sense of enjoyment from participating in environmental activities [33].

**Author Contributions:** Writing—original draft preparation, W.C.; writing—review and editing, W.C., P.M. and C.S. All authors have read and agreed to the published version of the manuscript.

**Funding:** This research received no external funding.

**Institutional Review Board Statement:** Ethical review and approval were waived for this study by the Institutional Review Board of Khon Kaen University.

**Informed Consent Statement:** Not applicable.

**Data Availability Statement:** Data available on request.

**Conflicts of Interest:** The authors declare no conflict of interest.

## Appendix A.

*Appendix A.1. Scenario 1: Non-Charity Running Event (SC1)*

"Let's say you have paid the application fee to run in the event 'WE CAN RUN: Running with the Sound of Khaen' to win a prize in Khon Kaen. It is a 10.5-km mini marathon, with the runners scheduled to be released at 6:00 am. The running route starts from the Si Than Gate of Khon Kaen University, running into the road in front of the Faculty of Law, around the central stadium, and through important places such as agricultural parks, Dean's buildings, White Bridge, Teak Road, and the Natural History Museum. Along the route, police monitor traffic and a team of pacers helps you reach the finish line within the intended time. Drinking water and first aid points are available throughout the distance. After the event, you will receive a thank you email and event pictures from the organizer."

*Appendix A.2. Scenario 2: Charity Running Event That Donated All the Proceeds from Registration Fees to a Charity (SC2)*

"Let's say you have paid the application fee to run in the event 'WE CAN RUN: Running with the Sound of Khaen' to fund the purchase of educational equipment for schools in rural areas that are in need. It is a 10.5-km mini marathon, with the runners scheduled to be released at 6:00 am. The running route starts from the Si Than Gate of Khon Kaen University, running into the road in front of the Faculty of Law, around the central stadium, and through important places such as agricultural parks, Dean's buildings, White Bridge, Teak Road, and the Natural History Museum. Along the route, police monitor traffic and a team of pacers helps you reach the finish line within the intended time. Drinking water and first aid points are available throughout the distance. After the event, you will receive a thank you email, event pictures, and the organizer also informs you that all the proceeds from application fees have already been given to schools in rural areas that are in need."

*Appendix A.3. Scenario 3: Charity Running Event That Donated a Portion of the Money from Registration Fees to a Charity (SC3)*

"Let's say you have paid the application fee to run in the event 'WE CAN RUN: Running with the Sound of Khaen' to fund the purchase of educational equipment for schools in rural areas that are in need. It is a 10.5-km mini marathon, with the runners scheduled to be released at 6:00 am. The running route starts from the Si Than Gate of Khon Kaen University, running into the road in front of the Faculty of Law, around the central stadium, and through important places such as agricultural parks, Dean's buildings, White Bridge, Teak Road, and the Natural History Museum. Along the route, police monitor traffic and a team of pacers helps you reach the finish line within the intended time. Drinking water and first aid points are available throughout the distance. After the event, you will receive a thank you email, event pictures, and the organizer also informs you that a portion of the proceeds from the application fees has already been given to schools in rural areas that are in need."

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
