# Peer review of "Do Charity or Non-Charity Sporting Events Have a Greater Influence on Participants’ Warm Glow?: An Experimental Survey"

_sustainability, doi:10.3390/su142416593_

Round 1

Reviewer 1 Report

Dear Authors,

Thank you for the interesting, informative, and valuable work that will contribute to a better understanding of the effects of participating in charity running events and altruism on warm glow, as well as the relationship between these variables. With the aim to improve the content of this research paper, I have some recommendations.

1.     As you mentioned in the Introduction chapter participating in charity running events might make participants feel a warm glow in addition to altruism. What else can you state as the aim of this study, besides understanding the factors influencing warm glow?

 2.     Since, people may be "impurely altruistic" in the warm-glow framework, which refers to the simultaneous maintenance of both altruistic and egoistic (selfish) incentives for giving, I recommend briefly mentioning these instances as well in Subheading 2.2

3.     In the Discussion section, it is also necessary to make clear what the organizers of such events need to focus on especially. How can the objectives of the organizer’s and the participants be reconciled? The authors are suggested to implement the aspect of warm glow discussion referring also to the general literature and to resume in a clear way the concluding aim of the article.

Reviewer 2 Report

ABSTRACT – I do not really understand the title of the study. Otherwise, very well written abstract

INTRODUCTION – could you discuss warm glow. What does that mean? Why is it important? Etc This introduction needs to give a bit of background to the study.

LITERATURE REVIEW – This section is overall well discussed, with suitable hypothesis

MATERIAL MTETHODS / RESULTS – I can’t discuss this section as I am not into quantitative analysis

MINOR CORRECTION!

Reviewer 3 Report

Thank you very much for the opportunity to read the paper. I think the authors have done pretty much, but the following points should be improved.

1. Clarify why they have selected Thai for this survey and used this regression model in this study.

2. The budget and time should not affect the validity of the results of any experiments and surveys. It's not a good idea to mention the budget constraints in the methods. Rather, the authors could mention the difficulties of data collection they faced during data collection. 

3. Data collection methods are not clear. How the authors distributed their questionnaire randomly to the park runners? As per my assumption, they might have gone to the parks, and whomever they could approach distributed it. Please clearly mention what you have done for data collection.

4. The Cronbach's Alpha is extremely high, which signifies that some items included in the model are redundant. I want the authors to confirm the results. If it is 0.98 it is extreme, send me the SPSS test results and questionnaire. Normally, an alpha over 0.9 is unacceptable.

5. Why the authors selected Thailand has no justification, and why not other parks? Why specifically that park has been selected for the data collection?

6.  What are the practical implications of this study?

Kumar
